# Leptin-Induced HLA-G Inhibits Myometrial Contraction and Differentiation

**DOI:** 10.3390/cells11060954

**Published:** 2022-03-10

**Authors:** Maeva Wendremaire, Tatiana E. Lopez, Marina Barrichon, Hang Zhang, Tarik Hadi, Xiang-Yang Ye, Fabrice Neiers, Marc Bardou, Paul Sagot, Carmen Garrido, Frédéric Lirussi

**Affiliations:** 1UMR 1231, Lipides Nutrition Cancer, INSERM, F-21000 Dijon, France; maeva.wendremaire@u-bourgogne.fr (M.W.); tatiana_lopez@etu.u-bourgogne.fr (T.E.L.); barrichon@hotmail.com (M.B.); zhanghanghznu@163.com (H.Z.); tarik.hadi21@gmail.com (T.H.); cgarrido@u-bourgogne.fr (C.G.); 2UFR des Sciences de Santé, Université Bourgogne Franche-Comté, F-25000 Besançon, France; fabrice.neiers@u-bourgogne.fr (F.N.); marc.bardou@u-bourgogne.fr (M.B.); 3School of Basic Medical Science, Hangzhou Normal University, Hangzhou 311121, China; 4School of Pharmacy, Hangzhou Normal University, Hangzhou 311121, China; xyye@hznu.edu.cn; 5Centre des Sciences du Goût et de l’Alimentation, INRAE, CNRS, Université Bourgogne Franche-Comté, F-21000 Dijon, France; 6CIC 1432, INSERM, Université de Bourgogne Franche-Comté, F-21000 Dijon, France; 7Service de Gynécologie-Obstétrique, Centre Hospitalo-Universitaire Dijon, F-21000 Dijon, France; paul.sagot@chu-dijon.fr; 8Department of Medical Oncology, Centre Georges François Leclerc, F-21000 Dijon, France; 9Plateforme PACE, Laboratoire de Pharmacologie-Toxicologie, Centre Hospitalo-Universitaire Besançon, F-25000 Besançon, France

**Keywords:** leptin, differentiation, labour, macrophage, myometrium, oxidative stress

## Abstract

Maternal obesity is associated with a wide spectrum of labour disorders, including preterm birth. Leptin, a pro-inflammatory adipokine and a key factor of obesity, is suspected to play a major role in these disorders. OB-R, its receptor, is expressed on macrophages and myocytes, two cell types critical for labour onset. Macrophages secrete reactive oxygen species/pro-inflammatory cytokines, responsible for myometrial differentiation while myocytes control uterine contractions. In this study, we assessed the effect of leptin on myometrial contraction and differentiation using our validated co-culture model of human primary macrophages and myocytes. We demonstrated that leptin had a different effect on myocytes and macrophages depending on the dose. A low leptin concentration induced a tocolytic effect by preventing myocytes’ contraction, differentiation, and macrophage-induced ROS production. Additionally, leptin led to an increase in HLA-G expression, suggesting that the tocolytic effect of leptin may be driven by HLA-G, a tolerogenic molecule. Finally, we observed that recombinant HLA-G also prevented LPS-induced ROS production by macrophages. Altogether, these data provide a putative molecular mechanism by which leptin may induce immune tolerance and therefore interfere with labour-associated mechanisms. Therefore, HLA-G represents a potential innovative therapeutic target in the pharmacological management of preterm labour.

## 1. Introduction

Obesity has become a worldwide epidemic and therefore a major public health issue [1]. Of particular concern is the increasing prevalence of obesity in pregnant women and women of reproductive age [2,3]. Indeed, obesity in pregnancy adversely impacts pregnancy outcomes [4] linked to a disruption in the parturition mechanisms leading to either preterm birth (PTB) [5,6,7] or delayed and difficult labour onset [8,9,10]. The influence of maternal obesity on labour remains controversial and a better understanding of the molecular events associated with obesity might help to understand the complexity of delivery outcomes.

Maternal obesity is associated with a low-grade inflammatory state characterised by a moderate but chronic systemic rise in a number of adipokines, including leptin [11]. Leptin—a 16-kDa protein encoded by the *LEP* gene—is secreted mainly by the adipose tissue and plays an integral role in food intake, satiety, and energy expenditure [12]. Plasma leptin levels were reported to correlate with both body mass index and the total amount of fat mass and are elevated in obese subjects leading to a state of leptin resistance [13,14]. Leptin concentration is increased during pregnancy [15], primarily due to leptin production by the placenta [16]. Although leptin is not secreted by the myometrium, its receptors OB-R—encoded by the *LEPR* gene—were identified in this tissue [17], which argues in favor of a paracrine role controlling myometrial function. Such an autocrine/paracrine effect has already been demonstrated during the placentation process. Indeed, leptin was shown to promote human leukocyte antigen-G (HLA-G) expression in placental trophoblasts, suggesting an important role in the regulation of immune mechanisms at the fetal–maternal interface [18].

Besides physiological effects, accumulating evidence suggests that leptin signalling might be involved in the pathophysiology of delivery in obese women. A study revealed that maternal *LEP* and *LEPR* polymorphisms were significantly associated with increased risk for PTB [19]. Indeed, women carrying *LEP* AA genotype had a significant 2.53-fold increased risk for PTB compared to other *LEP* genotypes. Similar results were observed with *LEPR* polymorphisms, where women carrying the AA and AG genotypes had a significant 4.32- and 4.76-fold increased risk for severe PTB compared to women carrying the GG genotype [19]. Contrariwise, in vitro studies have shown that leptin is able to oppose both spontaneous and oxytocin-induced contractions in human myometrium [20]. It also prevents myometrial cell apoptosis and extracellular matrix remodelling [17,21]—two common features of uterine preparation for labour—in an in vitro model of human myometrial inflammation. Recently, our team also demonstrated that leptin was able to induce myometrial smooth muscle cells proliferation [22]. Taken together, these results suggest that leptin may contribute to delayed labour onset as commonly observed in post-date pregnancies.

In labour, macrophages massively infiltrate myometrium and secrete pro-inflammatory cytokines and reactive oxygen species (ROS) inducing the expression of specific labour-associated markers [23,24,25]. Our team previously developed a co-culture model of myocytes and macrophages and demonstrated the interaction between these two cell types in lipopolysaccharide (LPS)-induced labour mechanisms [26]. Indeed, macrophages activated by LPS activate myocytes through ROS production, inducing myometrial differentiation and subsequent contractions.

Considering these data, we wanted to investigate the impact of leptin, at different concentrations, in the contraction and differentiation/activation steps of myometrial cells, to explore its implication in parturition related-disorders. We also attempted to determine whether leptin was involved in the interaction between macrophages and myometrial cells and whether this interaction involved the tolerogenic molecule HLA-G. In the present study, we identified leptin as a tocolytic agent and demonstrated that myometrial cells could modulate macrophage activation via leptin-induced expression of HLA-G, and thus oppose differentiation and contraction processes induced by macrophages.

## 2. Materials and Methods

### 2.1. Drugs and Solutions

Human leptin (L4146), LPS from *Escherichia coli* (serotype 055:B5; L2880), H_2_O_2_ (H1009), accutase (A6964), Lucifer Yellow (LY; L0259) solution, lucigenin (M8010) and β-NADPH (N1630) were obtained from Sigma-Aldrich (Merck, Darmstadt, Germany). Leptin receptor antagonist (super-active human leptin antagonist, SHLA; LAN-2 GENE ID 3952) was purchased from Protein Laboratories (Rehovot, Israel). GM-CSF (Granulocyte Macrophage-Colony Stimulating Factor; 130-095-372) was obtained from Miltenyi Biotec (Paris, France). Biotinylated anti-HLA-G antibody (MEM/G1) was purchased from Exbio (Vestec, Czech Republic) and HLA-G recombinant protein (HLA-G-001H) from Creative Biomart (Shirley, NY, USA). Rat tail type-I collagen solution, Alexa Fluor 555 phalloidin and HRP-streptavidin-conjugated antibody were obtained from ThermoFisher Scientific (Waltham, MA, USA). Anti-COX-2 (cyclooxygenase-2, sc-19999), anti-GAPDH (sc-25778) and HRP-conjugated anti-rabbit antibodies (sc-2004) were obtained from Santa Cruz Biotechnology (Santa Cruz, CA, USA). All drugs were dissolved according to the manufacturers’ instructions.

### 2.2. Cell Culture

#### 2.2.1. Myometrial Cells

Myometrial biopsies were collected from normal-weight women undergoing elective Caesarean section delivery, between 38 and 40 weeks of gestation, for cephalopelvic disproportion, before labour onset. Pre-pregnancy body mass index for the patients was between 18.5 and 24.9 kg/m^2^. This study was approved by the competent French authorities (Comité de Protection des Personnes and Agence Nationale de Sécurité du Médicament et des Produits de Santé (2012-A00136-37)). Written informed consent was obtained from all donors. Human primary myometrial cell lines were obtained as previously described [25], and cultured at 37 °C under 5% CO_2_ in Dulbecco’s modified Eagle’s medium (DMEM, Dominique Dutscher, Brumath, France), with 10% fetal bovine serum (FBS, Dominique Dutscher, Brumath, France) and 1% PSA (Penicillin G, Streptomycin, Amphotericin B, Dominique Dutscher, Brumath, France). Myometrial cells were used from the third to the sixth passage. Myocytes were seeded in 6-well plates for qRT-PCR, Western blotting, NADPHox activity and flow cytometry (10^6^ cells per well), in 24-well plates for fluorescence and Scrape loading/dye transfer assay (2.5 × 10^4^ cells on sterile glass coverslip and 2.5 × 10^5^ cells per well, respectively) and in 35 mm untreated suspension culture dish for collagen lattices (1.5 × 10^5^ per dish). Cells were left to adhere for 24 h and starved in serum-free DMEM for 48 h before co-culture and/or stimulation protocol.

#### 2.2.2. Monocytes/Macrophages

Human monocytes were isolated from peripheral blood mononuclear cells (PBMC) from buffy coats (Etablissement Français du Sang, Besançon, France) by two-step density gradient centrifugation, as previously described [27]. Monocytes were differentiated into macrophages with 100 ng/mL GM-CSF, at 37 °C under 5% CO_2_ for 7 days in Roswell Park Memorial Institute medium (RPMI, Dominique Dutscher, Brumath, France) with 10% FBS and 1% PSA. Cells were seeded in 6-well plates for NADPHox activity and flow cytometry (10^6^ cells per well), in 24-well plates for fluorescence (2.5 × 10^5^ cells on sterile glass coverslip). 

#### 2.2.3. Co-Cultures

For co-culture experiments, monocytes were seeded in 100 mm culture dishes (10 × 10^6^ cells). Then, differentiated macrophages were harvested using accutase and added to myocytes at a ratio of 1:4 (macrophage:myocyte) in serum-free medium. For collagen lattice experiments, myocytes and macrophages were seeded together.

### 2.3. Stimulation Protocol

Cells were stimulated either with leptin at two different concentrations (5 and 50 ng/mL), LPS (100 ng/mL) or H_2_O_2_ (1 µM) in serum-free medium. The specificity of the effect of leptin was assessed by using leptin receptor antagonist SHLA (100 ng/mL) added 1 h before leptin. To mimic HLA-G action, macrophages were stimulated with LPS plus recombinant HLA-G at 0.2 or 1 µg/mL. For each experiment, time-matched vehicle controls were performed.

### 2.4. Collagen Lattices

Myometrial cells with or without macrophages were included simultaneously in collagen gels as previously described [26]. The cells in the lattice were then stimulated with LPS +/− leptin or H_2_O_2_ +/− leptin. Images of the floating gels were captured before adding the test agents and then every day for up to 4 days and digitised using ChemidocTM XRS+ (BioRad, Hercules, CA, USA). The areas of the lattices were measured with Image J software (v1.48; National Institutes of Health, Bethesda, MD, USA), and the percentage of collagen contraction was calculated by comparing the 96 h surface area to the basal surface area.

### 2.5. Western Blotting

Cells were lysed in 100 µL of cold RIPA buffer. Total protein content was quantified by Nanodrop (ThermoFisher Scientific); samples were dissolved in Laemmli 4X buffer (BioRad) and boiled for 3 min at 95 °C. Thirty micrograms of total protein were loaded per well and subjected to 12% SDS-PAGE (sodium dodecyl sulfate–polyacrylamide gel electrophoresis) before being transferred to a 0.2 µm polyvinylidene fluoride membrane (Trans-Blot Turbo^TM^ Mini Transfer System, BioRad). Nonspecific antibody binding sites were blocked for 1 h in 5% non-fat dried milk powder in PBS with 0.1% Tween 20 (PBST) solution at room temperature (RT). Blots were probed at 4 °C overnight with rabbit anti-COX2 (1/500) or biotinylated anti-HLAG (1/1500) primary antibody in 1% non-fat dried milk powder in PBST. After three washes with PBST, blots were incubated for 1 h at RT with horseradish peroxidase-conjugated anti-rabbit antibody (1/5000) or HRP-streptavidin-conjugated antibody (1/4000) respectively. Immunoreactive proteins were detected by enhanced chemiluminescence (SuperSignal^TM^ West Dura Extended Duration Substrate, ThermoFisher Scientific) and the signal was detected by Chemidoc^TM^ XRS+ (BioRad). Densitometric analysis was performed using Image Lab^TM^ software (v5.2.1; BioRad). The blots were stripped with Re-Blot Plus-Strong (Millipore, Merck) and revealed with GAPDH antibody (1/1000). Results were expressed as mean +/− standard error of the mean (SEM) in arbitrary density units (ADU).

### 2.6. Quantitative Real-Time RT-PCR (qRT-PCR)

Total RNA was isolated using NucleoZOL (Macherey Nagel, Hoerdt, France) according to the manufacturer’s instructions. Total RNA content was determined by Nanodrop (ThermoFisher Scientific). The integrity of RNA was verified by optical density absorption ratios 260 nm/280 nm and 260 nm/230 nm ≥ 2. RNA samples (2 µg) were reverse transcribed using Maxima First Strand cDNA synthesis kit (K1642, ThermoFisher Scientific). Quantitative real-time PCR was performed on a StepOnePlus^TM^ system (Applied Biosystems, ThermoFisher Scientific) using the Maxima SYBR Green System (K0222, ThermoFisher Scientific) and 375 nM of the appropriate forward and reverse primers (*HLA-G* forward: 5′-TTGCTGGCCTGGTTGTCCTT-3′, reverse: 5′-TTGCCACTCAGTCCCACACAG-3′; *GAPDH* forward: 5′-GGAGTCAACGGATTTGGT-3′, reverse: 5′-GTGATGGGATTTCCATTGAT-3′). *GAPDH* gene was chosen as housekeeping gene as previously described [28]. All primer pairs were purchased from Merck. The following PCR program was used: 10 min at 95 °C, 40 cycles of 15 s at 95 °C and 60 s at 60 °C. A melting curve analysis was performed which consisted of 70 cycles of 15 s with a temperature increment of 0.5 °C/cycle starting at 60 °C. Cycle threshold (Ct) analysis was performed with StepOne^TM^ software (v2.3; Applied Biosystems, ThermoFisher Scientific). Relative expression (RE) was calculated as follows: RE = 2^−ΔCt^, where Ct = Ct _HLA-G_ − Ct _GAPDH_, and presented with the control condition set as one.

### 2.7. Fluorescence Analysis

Cells were plated on coverslips in 24-well dishes and subjected to the different stimulation protocols. For DHE (dihydroethidium) staining, cells were cultured for a further 30 min at 37 °C and 5% CO_2_ after stimulation, in a PBS solution containing 10 μM DHE, then washed and fixed in 4% paraformaldehyde (PFA) for 5 min at 4 °C [25]. For phalloidin staining, after 24 h of treatment, cells were fixed in 4% PFA for 5 min at 4 °C and permeabilised with 0.1% saponin in PBST for 10 min at RT. Filamentous actin (F-actin) was stained with Alexa Fluor 555 phalloidin (1/80) in 0.1% saponin in PBST for 30 min at RT, protected from light. After two washes with PBS, coverslips were mounted onto superfrost slides in a ProLong^®^ Gold Antifade Mountant with DAPI (4–6 diamidino-2-phenylindole; Molecular Probes, Leiden, The Netherlands). Slides were viewed on an epifluorescence microscope Eclipse E400 (Nikon, Tokyo, Japan). Five representative pictures were taken in randomly selected fields for each labelling. Phalloidin and DHE staining were measured with mean fluorescence intensity by Image J software (v1.48; National Institutes of Health, Bethesda, MD, USA). Results were expressed as mean +/− SEM.

### 2.8. Scrape Loading/Dye Transfer Assay

Scrape loading/dye transfer method (SL/DT) was used to analyse gap junction activity of cells, as previously reported [29]. Myocytes and macrophages were seeded in 24-well plates and submitted to stimulation. Cell layer was cut with a scalpel and then incubated with LY (0.05%) for 3 min at RT. Cells were rinsed three times with PBS. LY fluorescence was visualised with an inverted fluorescence microscope (Station Cell Observer, Carl Zeiss Microscopy GmbH, Jena, Germany). The distances of Lucifer Yellow diffusion after scrape loading were compared between the differently treated groups. Three representative pictures were taken for each experiment. Pictures were analysed and merged using Image J software (v1.48; National Institutes of Health, Bethesda, MD, USA).

### 2.9. Flow Cytometry

After stimulation, cells were cultured for 30 min, at 37 °C and 5% CO_2_, in DHE (10 µM in PBS), scraped out, and centrifuged (10 min, 1500 rpm, 4 °C). Cells were fixed for 5 min in 4% PFA and analysed using an LSRII flow cytometer (Becton Dickinson, San Diego, CA, USA). Primary Size–Granularity dot plot allowed to discriminate cells from debris, and DHE-positive cells were obtained by comparing fluorescence at 575 nm versus unstained samples, using Flowjo software (v10.8.1; Becton Dickinson) [25]. Results were expressed as mean +/− SEM.

### 2.10. NADPHox Activity

Treated cells were incubated in 100 µL of ice-cold lysis buffer (50 mM TrisHCl pH 8, 1 mM EDTA, 150 mM NaCl, 1% NP40 with protease and phosphatase inhibitors) for 30 min and scraped out. Cell lysate (50 µL) was transferred into 96-well plate. The protocol was adapted from previous publications [25,30]. Baseline activity was measured for 20 min after the addition of 20 µM lucigenin. Cells were then stimulated by the addition of 200 µM β-NADPH, and active levels were measured for 2 h using EnVision photomultiplier (PerkinElmer, Zaventem, Belgium) measuring between 390 and 620 nm. Baseline activity was subtracted, and NADPHox adjusted activity was normalised to protein concentration. Total protein content was quantified by Nanodrop (ThermoFisher Scientific).

### 2.11. Statistical Analysis

The normality of datasets was determined by the Shapiro–Wilk test. The homogeneity of variances was determined by the Brown-Forsythe test. The differences between more than two groups were determined by one-way ANOVA followed by Bonferroni’s test, for normally distributed data. The Kruskal-Wallis test followed by Dunn’s multiple comparison test were utilised for non-parametric data. The Friedman test was used for one-way repeated measures analysis of variance by ranks. Statistical analysis was performed with GraphPad Instat software (v7; San Diego, CA, USA). Statistical significance between groups was defined as *p* < 0.05 (*), *p* < 0.01 (**) and *p* < 0.001 (***) versus Control conditions and *p* < 0.05 (#), *p* < 0.01 (##) and *p* < 0.001 (###) versus LPS conditions.

## 3. Results

### 3.1. A Low Dose (5 ng/mL) of Leptin Can Prevent Myometrial Contraction Induced by LPS Stimulation

We first assessed the effect of leptin on myocytes contraction alone using collagen lattices. We did not observe any difference compared to the control until 96 h after the leptin stimulation (Appendix A). We also studied whether leptin was able to induce myometrial cell contractions in our validated co-culture model [26]. When myometrial cells co-cultured with macrophages were stimulated with leptin without LPS, we observed that a high concentration of leptin (50 ng/mL) induced spontaneous myocyte contractions (Appendix A). Next, we assessed whether leptin was able to block LPS-induced contractions of myometrial cells when cultured with macrophages. When LPS was added, LPS stimulation led to an increase in spontaneous contractions that was prevented by the addition of leptin at the concentration of 5 ng/mL (Figure 1A,B). SHLA, a selective antagonist of OB-R, abrogated a leptin-induced decrease in contractions, indicating that this effect selectively depended on leptin receptor stimulation (Figure 1A,C). Finally, we wanted to know if the effect of leptin on inflammation-induced contractions could be caused by a direct effect on prostaglandin synthesis, which is the endogenous triggering agent of contractions. For that, we studied the expression of COX-2, one of the main enzymes participating in prostaglandin synthesis. We observed that leptin at the concentration of 5 ng/mL prevented LPS-induced COX-2 expression (Figure 1D,F) and that this effect was lost upon the addition of SHLA (Figure 1E,G).

### 3.2. A Low Dose (5 ng/mL) of Leptin Prevents the Different Stages of Myometrial Differentiation Induced by LPS

Since myometrial cell activation/differentiation precedes contraction, we investigated the potential implication of leptin in the differentiation process: cytoskeleton reorganisation and functional gap junctions (GJ) formation, required to induce myometrial contractions. We first investigated the ability of leptin to interfere with LPS-induced actin fibers as an indicator of cytoskeleton reorganisation. LPS stimulation modified myometrial morphology in a triangular-shaped cell, representative of a labouring myocyte profile. The addition of leptin at 5 ng/mL to LPS completely prevented this reorganisation and the actin fibers remained parallel along the length of the cells (Figure 2A). Once again, we observed that this effect in the cytoskeleton reorganisation was lost upon the addition of SHLA. Then, we evaluated the formation of functional GJ between myocytes, a critical step for effective and synchronised contractions. We studied gap junction intercellular communication (GJIC) function by the SL/DT assay using the GJ permeable fluorescent dye LY. We observed that leptin prevented LPS-induced LY transfer to adjacent cells via open GJ channels and remained confined at the wound edge of the cells in contact with the dye during the scrape–loading process (Figure 2B). This effect was lost in the presence of SHLA, a specific antagonist of the leptin receptor. 

Taken together, these results indicate that a low leptin dose can prevent myometrial contractions, differentiation, and/or activation induced in an inflammatory context and that these effects are selectively mediated by leptin receptor stimulation.

### 3.3. A Low Dose (5 ng/mL) of Leptin Prevents ROS Production in Co-Cultured Cells, but Not in Macrophages

In the second set of experiments, we investigated which cell type was responsible for the observed effect of leptin. Indeed, the leptin receptor OB-R is expressed in both human macrophages and myometrial cells. Furthermore, we and others have demonstrated that macrophages were the main source of ROS production in the human myometrium and a key cell type involved in the onset of inflammation-induced preterm labour [23,25,26,31,32]. To assess whether leptin could block the oxidative burst responsible for myometrial cells activation, cultures of macrophages and myocytes were treated with LPS in the presence or absence of leptin. As shown in Figure 3, a low concentration of leptin (5 ng/mL) blocked LPS-induced NADPHox activity (Figure 3A) and the subsequent ROS production. This latter effect was highlighted by the decrease in DHE-positive cells by flow cytometry (Figure 3B) and DHE staining by fluorescence (Figure 3C,D), whereas leptin had no effect on ROS production without LPS stimulation (Appendix A). These data confirm that low doses of leptin can mitigate an inflammatory signal in the co-culture of myocytes and macrophages. 

Since macrophages are the major contributors to oxidative stress, we investigated the effect of leptin on ROS production in macrophages. In response to LPS stimulation, leptin failed to prevent NADPHox activity (Figure 4A) and ROS production (Figure 4B–D). 

However, when examining the effect of leptin alone on macrophages, leptin stimulation at 50 ng/mL induced ROS production (Appendix A). Taken together, these results suggest that the anti-oxidant effect of leptin on co-culture is not mediated by a direct action on macrophages.

### 3.4. The Anti-Oxidant Effect of Leptin Is Mediated by Myometrial Cells through HLA-G Expression

To explore the role of leptin on myometrial cells, we stimulated myocytes with leptin and H_2_O_2_, which mimics the oxidative stress generated by macrophages stimulated with LPS. H_2_O_2_ stimulation induced the contraction of the collagen lattice when myometrial cells were cultured alone, which was fully abrogated by the addition of leptin at 5 ng/mL (Figure 5A,B). Figure 5C shows that the effect of leptin on myocyte contraction induced by H_2_O_2_ started 24 h post-stimulation. This result indicates that the effect of a low leptin dose on contraction is caused by a direct effect on myocytes.

In the last set of experiments, we considered HLA-G as one of the mediators of the effect of leptin, as previously reported in trophoblasts [18]. A time-course stimulation with leptin on myometrial cells demonstrated that the mRNA levels of *HLA-G* were increased after 8 h and 24 h of stimulation with both concentrations of leptin, 5 ng/mL and 50 ng/mL (Figure 6A). We next confirmed at the protein level this increase in HLA-G expression after 24 h and 48 h stimulations (Figure 6B,C). Furthermore, we observed an increase in *HLA-G* mRNA expression induced by 5 ng/mL of leptin in response to H_2_O_2_ stimulation (Figure 6D). 

Finally, macrophages were stimulated by LPS and recombinant HLA-G at two concentrations (0.2 and 1 µg/mL). We observed that stimulation with recombinant HLA-G prevented LPS-induced NADPHox activity (Figure 7A). The subsequent release of ROS was also prevented, as shown by the decrease in the percentage of DHE-positive cells revealed by FACS (Figure 7B) and DHE staining (Figure 7C,D).

Taken together, these results imply that leptin induces HLA-G expression in myocytes regardless of the leptin dose and that HLA-G inhibits LPS-induction of ROS production in macrophages.

## 4. Discussion

This paper focused on controversial clinical observations regarding the labour outcomes of obese women, whether they experienced preterm birth or post-date pregnancy [5,6,7,8,9,10]. To do so, we investigated the effect of leptin on the pathophysiology mechanisms observed on an LPS-induced model of PTL, using a reported co-culture model of human primary myocytes and macrophages [26], in which cells were stimulated with LPS to mimic inflammation and leptin as a feature of obesity. Indeed, leptin is mainly secreted by the adipose tissue and plasma leptin levels were reported to be strongly correlated with both body mass index and the total amount of fat mass. Regarding the literature, plasma leptin concentrations were defined around 5–15 ng/mL and 40–60 ng/mL, respectively, for a lean and an obese adult woman [13]. Thus, we chose leptin doses at 5 ng/mL for physiological and 50 ng/mL for obese conditions. Even if it is difficult to relate plasma concentration to a myometrial cellular exposure, this ratio of 1:10 between the doses is preserved.

We demonstrated that a low dose of leptin (5 ng/mL) exhibits a tocolytic effect by preventing myometrial cells contraction and differentiation (cytoskeleton reorganisation and cell synchronisation). These effects were specific to OB-R since they were prevented by SHLA, a selective OB-R antagonist. However, this tocolytic action of leptin is only observed in response to an inflammatory stimulus, since a low leptin level had no effect on myometrial cells in the absence of LPS stimulation. These data strengthen the need to work with co-cultures of myocytes and macrophages and therefore support the interest of our in vitro model. These data also corroborate previous results showing an inhibitory effect of leptin on myometrium biopsies [17,20,21]. Actually, leptin was shown to inhibit spontaneous and oxytocin-stimulated myometrial contractions and disrupt collagen degradation by metalloproteinases and cervical cell apoptosis [17,20,21]. Altogether, these data strongly suggest considering leptin as a tocolytic agent in women with preterm labour, taking into account the dose used [33]. 

Contrariwise, this tocolytic effect is no longer observed with a high leptin concentration (50 ng/mL), representative of obese women’s conditions. We even observed that a high concentration of leptin induces spontaneous contractions and pro-oxidant effects, without any LPS stimulus. This is consistent with the results observed in clinical studies showing a link between obesity/leptin and PTB [5,6,7,19]. Maternal *LEP* and *LEPR* polymorphisms were significantly associated with increased risk for PTB [19] and additionally, in women in active labour compared to those who were not in labour or before labour induction [34,35].

This versatile effect of leptin may be explained by the loss of specificity of leptin for its receptor at high concentrations. Indeed, we previously demonstrated that the leptin effect was no longer specific for the leptin receptor at a concentration of 50 ng/mL and was able to bind to the IL-6 receptor (IL-6R) [22]. Furthermore, in addition to the three classically described signalling pathways—JAK/STAT, PI3K/Akt, and ERK1/2 pathways—leptin was also shown to activate the IL-6 pathway. Thus, the involvement of the IL-6/IL-6R axis leads to modification of leptin receptor signalling pathways and allows the activation of the nuclear factor-kappa B (NF-κB) signalling pathway [22], a key regulator of the terminal processes of human labour and delivery that drives the expression of many pro-labour genes [36,37]. In line with our previous work, our leptin dose–response provides some mechanistic answers that could partially explain the discrepant results on labour observed with epidemiological studies.

The second major finding of the paper concerns leptin’s effect on oxidative balance. Indeed, we surprisingly observed that leptin did not prevent inflammation-induced ROS by macrophages when cultured alone, unlike it does when co-cultured with myocytes. As a pro-inflammatory cytokine, leptin acts as a growth factor for the monocytes, promoting phagocytic function and proliferation of circulating monocytes, inducing the production of pro-inflammatory cytokines (TNF-α (Tumor Necrosis Factor), IL-6 and IL-12) and stimulating the oxidative burst, as well as the chemotactic responses mediating the inflammatory infiltrate [38]. Moreover, in vivo experiments showed that leptin may enhance antitumor effects as it increased M1-like tumor-associated macrophage frequency compared with non-leptin-treated mice [39]. Nevertheless, leptin was also reported to promote immune tolerance on several cell types and immune escape mechanisms in certain types of tumours [18,40]. Unfortunately, mechanistic data about leptin-induced tolerance are poorly documented. Indeed, only one study conducted on trophoblasts during placentation reported this tolerance and implicated HLA-G as a mediator of these effects [18]. HLA-G is a non-classical MHC class I molecule that exerts important tolerogenic functions by inhibiting different immune-competent cells, such as lymphocytes, monocytes, macrophages, monocytes, and NK cells. HLA-G is a protein expressed as membrane-bound molecules (HLA-G) or as soluble isoforms (s-HLA-G) [41].

Given the well-known immunosuppressive properties of HLA-G, we decided to evaluate its expression on myometrial cells in response to leptin +/− H_2_O_2_ and its effects on LPS-induced ROS secretion by macrophages. We observed that both concentrations of leptin (5 and 50 ng/mL) induced an increase in HLA-G expression exclusively by myocytes. Moreover, an increase in *HLA-G* expression induced by leptin is also noticeable following stimulation with H_2_O_2_, a terminal metabolite of oxidative stress, suggesting that the tocolytic effect of leptin is driven by HLA-G. Contrary to our results, Zhou et al. [42] have shown that high levels of H_2_O_2_ down-regulate HLA-G expression in placental trophoblasts during pre-eclampsia. These divergent results may be explained by the high concentration of leptin found in pre-eclamptic patients [43]. Additionally, to further investigate the effects of HLA-G on oxidative balance, we demonstrated that recombinant HLA-G prevents LPS-induced ROS production by macrophages. To our knowledge, this is the first study that explored the effects of HLA-G on oxidative stress production. Only the effects of HLA-G on oxidative stress damage were investigated in a trophoblast model of pre-eclampsia and demonstrated that trophoblasts expressing HLA-G were more vulnerable to oxidative stress [42]. Moreover, recombinant sHLA-G drove the differentiation of macrophages with “immuno-modulatory” characteristics, including reduced expression of the M1 macrophage marker CD86 and increased expression of the M2 macrophage marker CD163 [44]. Both oxidative stress and pro-inflammatory cytokines are micro-environmental factors that can affect HLA-G expression and, therefore, its biological function.

HLA-G appears to be especially relevant during pregnancy since it was reported to be involved in implantation and placentation processes. Decreased levels of sHLA-G were found to be related to embryo implantation failure, recurrent spontaneous abortion, placental abruption, and pre-eclampsia [45]. Among obese pregnant women, maternal concentrations of sHLA-G increased from the first to the second trimester and before delivery, while in healthy controls, sHLA-G levels gradually decline as the pregnancy advances [46,47]. Therefore, it could be interesting to relate sHLA-G plasma concentrations to the risk of developing a particular pathology during pregnancy or parturition. sHLA-G measurement could be of special interest in clinical trials including obese women with dysfunctional labour. Taken together, the results of the present study underline the importance of HLA-G in the regulation of the activation of macrophages and therefore of the mechanisms involved in labour onset.

Finally, our results bring new insights into cellular cross-talk between macrophages and myocytes, two main effectors of labour onset. Preterm labour is well known to be associated with a massive macrophages infiltration into the myometrium that subsequently produces pro-inflammatory cytokines and induces contractions [23,24,25]. Thus, the effects of macrophages on myometrial cells differentiation/activation and labour-associated mechanisms have been well documented [24,25,26,48]. Nevertheless, few studies have investigated the interaction of myometrial cells with macrophages. A recent study demonstrated the ability of a broad-spectrum chemokine inhibitor to act on myometrial cells and prevent myocyte–macrophage interaction and ultimately myometrial contractility. This effect was mediated by a decrease in both chemokine secretion and *CAP* (contraction-associated proteins) gene expression by myometrial cells [49].

In this work, we demonstrated that myometrial cells could modulate macrophage activation by expressing HLA-G, and thus oppose differentiation and contraction processes induced by macrophages. More precisely, a low dose of leptin prevented LPS-induced ROS production by macrophages, through induction of HLA-G expression in myocytes. Although high leptin concentration increased myometrial HLA-G expression, it did not decrease ROS production in the co-culture. This discrepancy may be caused by the imbalance provoked by leptin at 50 ng/mL by inducing oxidative stress (pro-contractile) on macrophages and HLA-G expression (anti-contractile) on myometrial cells. In contrast, leptin, at 5 ng/mL, only induced HLA-G by myocytes but not ROS production by macrophages. As discussed previously, data concerning HLA-G expression in the utero-placental sphere are sparse. Nevertheless, given that leptin is not secreted by the myometrium but the placenta, these observations point out to a paracrine role of this hormone as a mediator controlling HLA-G expression in myometrial cells. The increased expression of HLA-G in myometrial cells is consistent with the tocolytic effects observed previously, thus preventing the secretion of ROS by activated macrophages. Consistent with this assumption, Barrientos et al. [18] showed that leptin acts as a paracrine/autocrine signal enhancing trophoblast HLA-G expression during placentation. In light of these results, maternal/placental leptin appears as an endogenous component of the differentiation machinery of cells from the utero-placental sphere, conferring them a tolerogenic phenotype to prevent oxidative and immunological damage. Taken together, this study brings new insights into the uterine redox and inflammatory biology involved in labour-associated processes. To our knowledge, our data are the first evidence that myometrial cells can interact with activated macrophages and underscore the importance of using cellular co-culture models to study the pathophysiological mechanisms of PTL.

## Figures and Tables

**Figure 1 cells-11-00954-f001:**
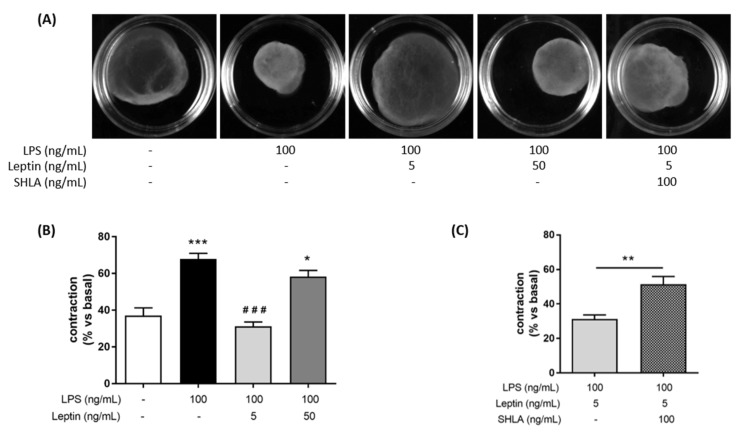
Tocolytic effect of leptin at 5 ng/mL on myocytes/macrophages co-culture in response to an LPS-stimulation. (**A**) Photographs of collagen lattices of co-cultures of myocytes and macrophages with or without of LPS, leptin, SHLA, for 96 h. Images are representative of six independent experiments. (**B**,**C**) Percentage contraction of collagen lattices (mean ± SEM) after 96 h of stimulation with LPS, leptin and/or SHLA, compared to the initial surface area, *n* = 6, * *p* < 0.05 and *** *p* < 0.001 versus Untreated, ^###^
*p* < 0.001 versus LPS. (**D**,**E**) Representative blots of COX-2 and GAPDH proteins stimulated with LPS, leptin and/or SHLA. (**F**,**G**) COX-2 and GAPDH protein relative content represented as fold induction (mean ± SEM) and expressed in ADU, *n* = 3–4, * *p* < 0.05 and ** *p* < 0.01 versus Untreated, ^#^
*p* < 0.05 versus LPS.

**Figure 2 cells-11-00954-f002:**
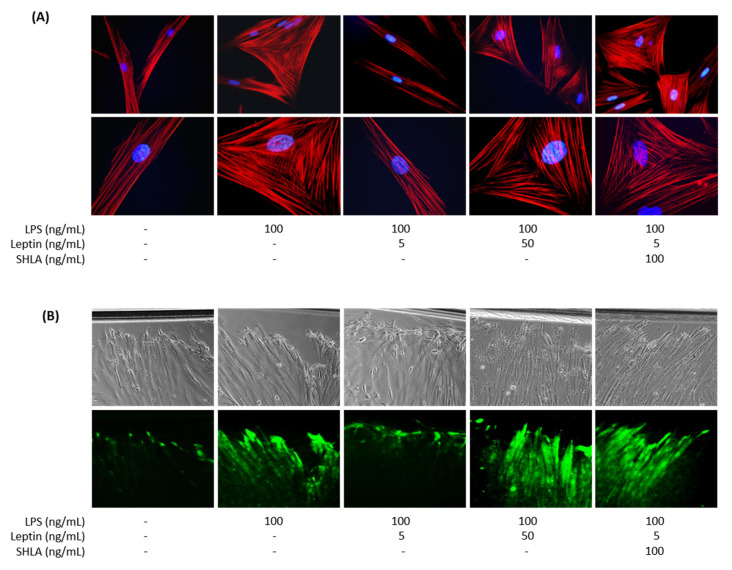
Effect of leptin on LPS-induced myometrial differentiation on myocytes/macrophages co-culture. Myocytes and macrophages were co-cultured and treated with LPS, leptin, and/or SHLA. (**A**) Merged fluorescence images of phalloidin (red) for actin cytoskeleton and DAPI (blue) for nucleus. Pictures were taken with an epifluorescence microscope at ×400 (upper row) and ×1000 magnification (lower row) in randomly selected fields and are representative of five pictures for each condition from six independent experiments. (**B**) Phase-contrast images (upper row) and fluorescence images of LY (lower row) transfer to adjacent cells after scrape loading. Pictures are representative of three independent experiments and were taken with an inverted microscope at ×200 magnification in randomly selected fields.

**Figure 3 cells-11-00954-f003:**
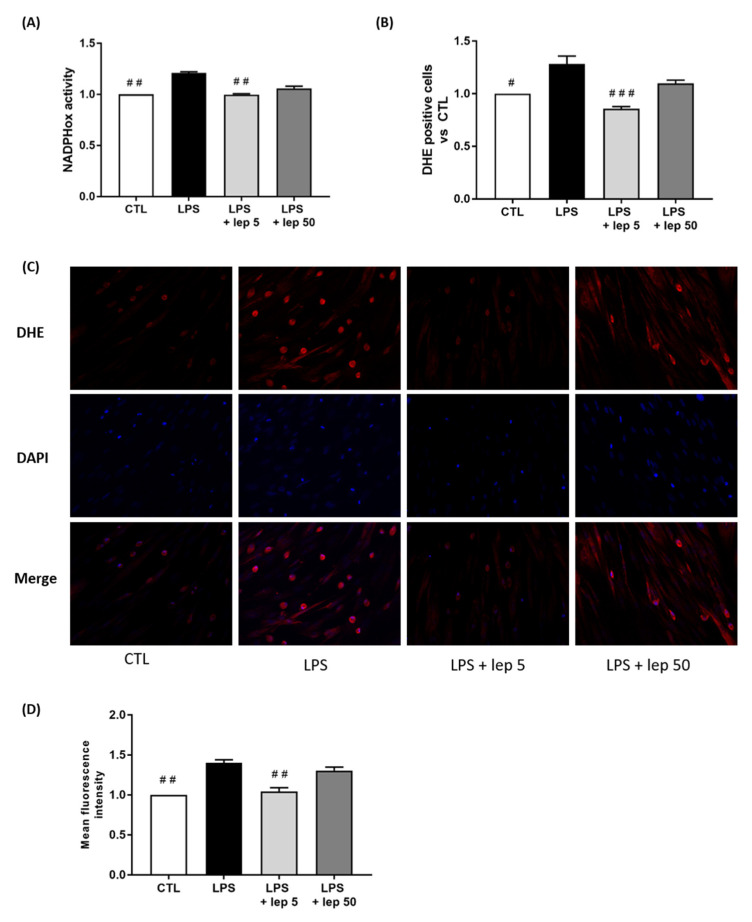
Effect of leptin on ROS production in co-cultured cells. Myocytes and macrophages were co-cultured and treated with LPS and leptin at 5 or 50 ng/mL. (**A**) NADPHox activity measured by luminometric analysis and related to the protein levels. Relative values versus CTL are represented as mean ± SEM, *n* = 4, ## *p* < 0.01 versus LPS. (**B**) Proportions of DHE-positive cells represented as fold induction versus CTL (mean ± SEM), *n* = 4, # *p* < 0.05 and ### *p* < 0.001 versus LPS. (**C**) Fluorescence images of DHE (red) and DAPI (blue) staining, taken with an epifluorescence microscope at ×200 magnification in randomly selected fields, representative of five pictures taken for each condition, *n* = 5. (**D**) Mean fluorescent intensity of DHE staining represented as fold induction versus CTL (mean ± SEM), *n* = 5, ## *p* < 0.01 versus LPS.

**Figure 4 cells-11-00954-f004:**
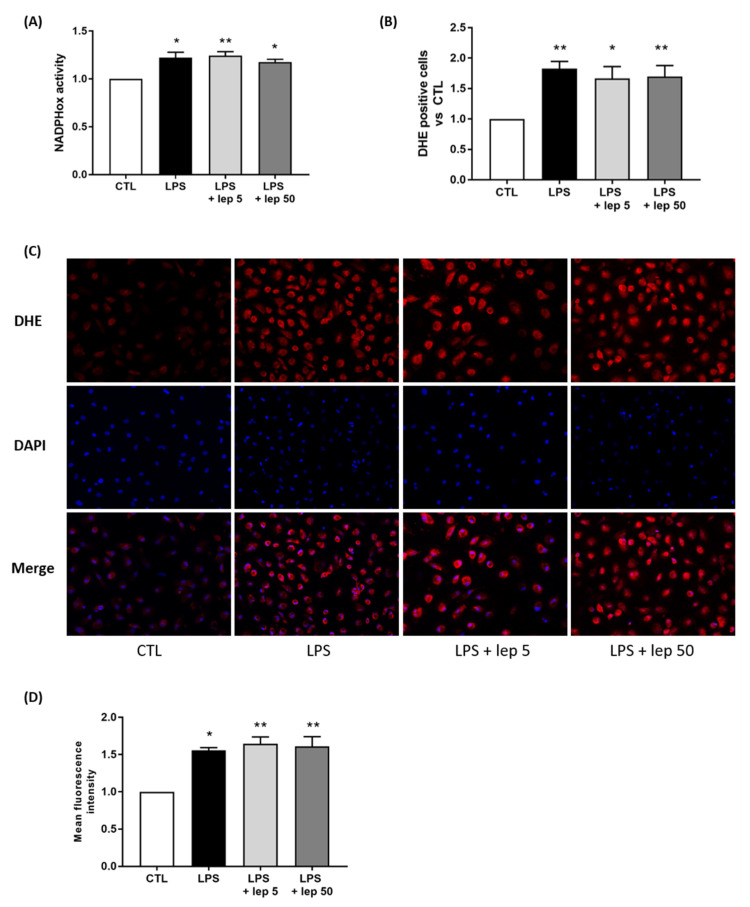
Effect of leptin on ROS production in macrophages. Macrophages were stimulated with LPS and leptin at 5 or 50 ng/mL. (**A**) NADPHox activity measured by luminometric analysis and related to the protein levels. Relative values versus CTL are represented as mean ± SEM, *n* = 8, * *p* < 0.05 and ** *p* < 0.01 versus CTL. (**B**) Proportions of DHE-positive cells represented as fold induction versus CTL (mean ± SEM), *n* = 5, * *p* < 0.05 and ** *p* < 0.01 versus CTL. (**C**) Fluorescence images of DHE (red) and DAPI (blue) staining, taken with an epifluorescence microscope at ×200 magnification in randomly selected fields, representative of five pictures taken for each condition, *n* = 5. (**D**) Mean fluorescent intensity of DHE staining represented as fold induction versus CTL (mean ± SEM), *n* = 5, * *p* < 0.05 and ** *p* < 0.01 versus CTL.

**Figure 5 cells-11-00954-f005:**
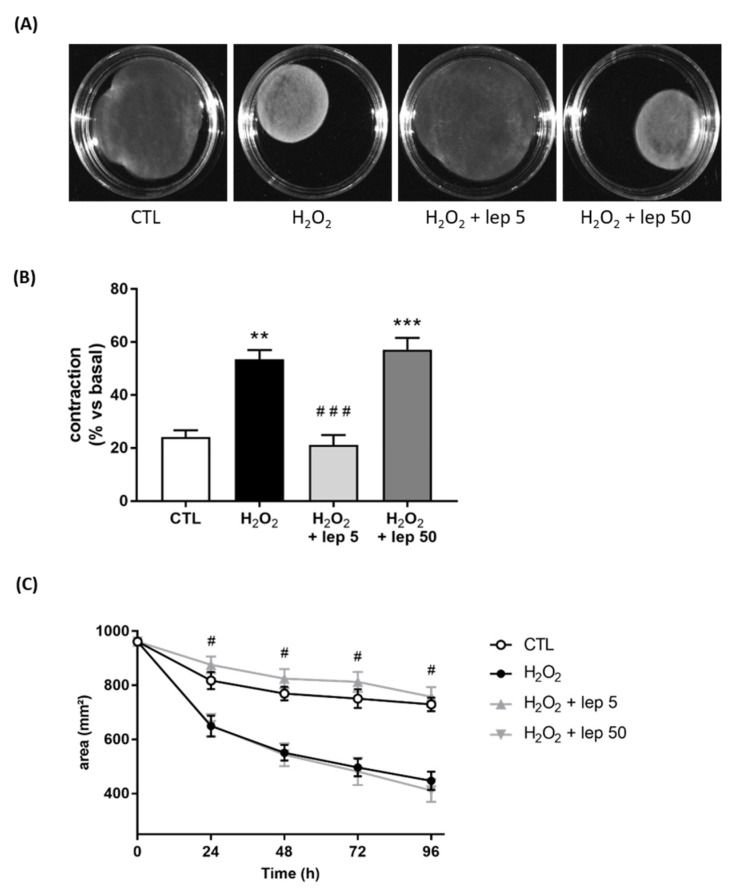
Effect of leptin on H_2_O_2_-induced contraction in myometrial cells. Myocytes were treated with or without H_2_O_2_ and leptin at 5 or 50 ng/mL. (**A**) Images of collagen lattices in presence or absence of H_2_O_2_ and leptin, for 96 h. Pictures are representative of four experiments. (**B**) Percentage contraction of collagen lattices (mean ± SEM) after 96 h of stimulation with H_2_O_2_ and leptin, compared to the initial surface area, *n* = 4, ** *p* < 0.01 and *** *p* < 0.001 versus CTL, ### *p* < 0.001 versus H_2_O_2_. (**C**) Time-course variation of collagen lattices area ± SEM up to 96 h of treatment with H_2_O_2_ and leptin. # *p* < 0.05 versus H_2_O_2_.

**Figure 6 cells-11-00954-f006:**
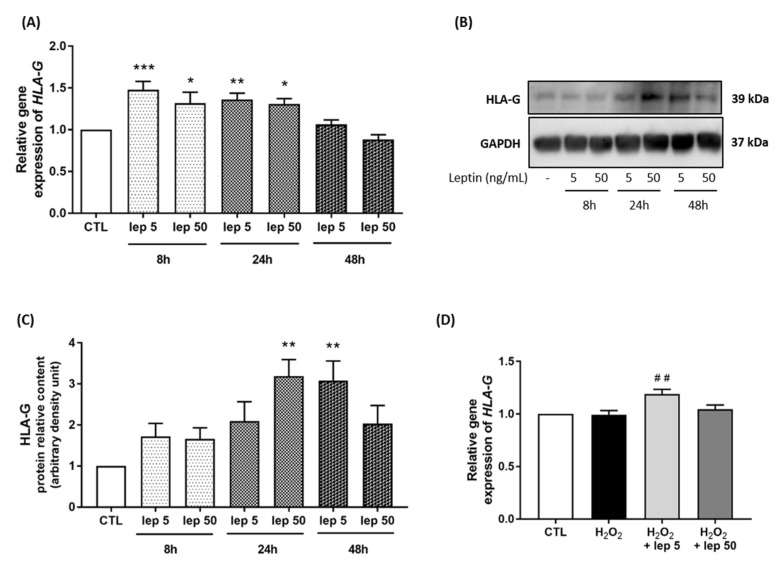
HLA-G expression induced by leptin in myometrial cells. Myocytes were stimulated with leptin at 5 and 50 ng/mL for 8 h, 24 h and 48 h. (**A**) Quantification of *HLA-G* mRNA levels analysed by qRT-PCR in myometrial cells. Results are expressed as fold induction versus CTL (mean ± SEM), of four replicate reactions. * *p* < 0.05, ** *p* < 0.01 and *** *p* < 0.001 versus CTL. (**B**,**C**) Representative blots of HLA-G and GAPDH proteins and densitometric analysis of HLA-G expression as fold induction versus CTL (mean ± SEM) and expressed in ADU, *n* = 4, ** *p* < 0.01 versus CTL. (**D**) Quantification of *HLA-G* mRNA expression in myometrial cells following H_2_O_2_ +/− leptin stimulation. ## *p* < 0.01 versus H_2_O_2_.

**Figure 7 cells-11-00954-f007:**
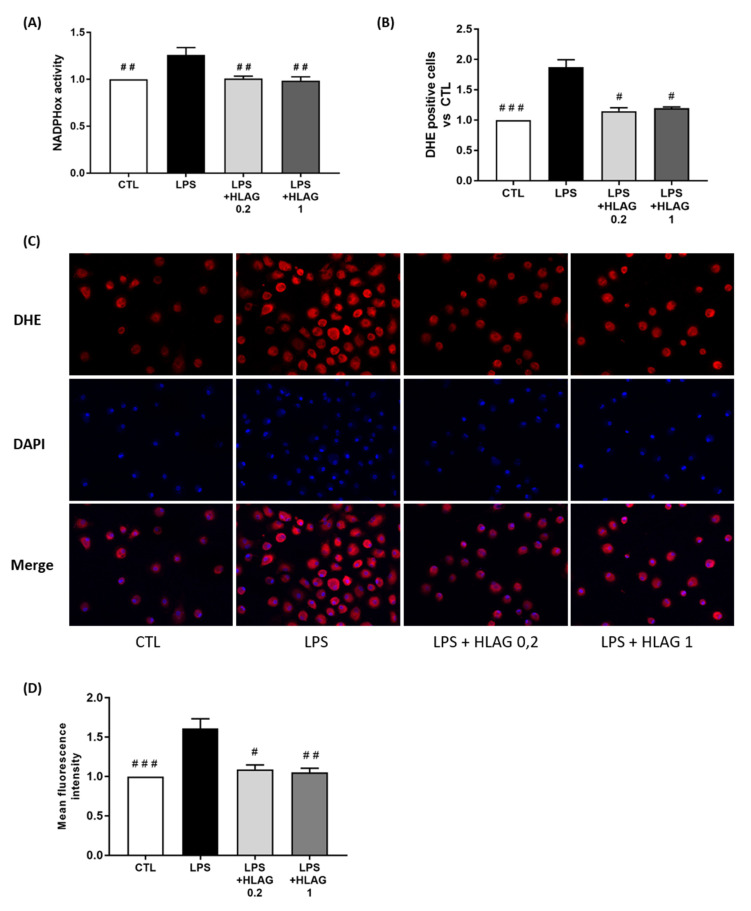
Effect of recombinant HLA-G on ROS production in macrophages. Macrophages were stimulated with LPS plus recombinant HLA-G at 0.2 or 1 µg/mL. (**A**) NADPHox activity measured by luminometric analysis and related to the protein levels. Relative values versus CTL are represented as mean ± SEM, *n* = 8, ## *p* < 0.01 versus LPS. (**B**) Proportions of DHE-positive cells represented as fold induction versus CTL (mean ± SEM), *n* = 6, # *p* < 0.05 and ### *p* < 0.001 versus LPS. (**C**) Fluorescence images of DHE (red) and DAPI (blue) staining, taken with an epifluorescence microscope at ×200 magnification in randomly selected fields, representative of five pictures taken for each condition, *n* = 8. (**D**) Mean fluorescent intensity of DHE staining represented as fold induction versus CTL (mean ± SEM), *n* = 8, # *p* < 0.05, ## *p* < 0.01 and ### *p* < 0.001 versus LPS.

## Data Availability

Not applicable.

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
