# Peer review of "Leptin-Induced HLA-G Inhibits Myometrial Contraction and Differentiation"

_cells, 2022, doi:10.3390/cells11060954_

Round 1
Reviewer 1 Report
Dear Authors,
I presented all my suggestions in the additional file.
With best regards.

Reviewer 2 Report
This is a very interesting article aimed to investigate the in vitro effects of leptin on the differentiation and contractility of human myometrial cells as well as the macrophage-induced ROS production. The main results are: leptin at low concentration inhibits contractility and differentiation of myocytes but not LPS-induced production of ROS by cocultured macrophages. Moreover, the tocolytic effect of leptin could be mediated by tolerogenic HLA-G. Taken together, these findings support the concept that leptin is involved in the control of parturition and could represent a potential target in future treatment of PTL.
The data are new and potentially relevant. The article is well written in all its parts. The methods are appropriate and multiple different methodologies have been used to strenghten the findings.
I have few questions:
- Did the authors test the effects of HLA-G on the contractility of myometrial cells in their in vitro system?
- To what extent the culture system used can be applied to myometrial contractility in vivo or in other in vitro systems (i.e., myometrial strips)?
- Is there any informaton on circulating levels of leptin in the setting of PTL as wella as throught normal pregnancy and labor?
- Is there a reason for which the experiments reported in Fig.1 do not include also any proof carried out by using LPS 100 ng/ml + Leptin 50 ng/ml + SHLA 100 ng/ml?
Reviewer 3 Report
This article examines the relationship between obesity, macrophages, and the effect of leptin on the contraction of smooth myocytes. The authors used a number of modern methods and obtained a number of facts showing that leptin prevents the contraction of smooth myocytes, their differentiation, and the release of ROS by macrophages.
There were comments on the methodological part of the work.
1. The authors demonstrate the results of real-time PCR, the graphs are presented in such a way that the control is taken as 1. This must be indicated in the materials and methods.
2. To establish statistically significant differences, the authors used one-way analysis of variance, as well as the Bonferroni test, which require a normal distribution of the studied characteristics. It is hard to imagine that such a rule was always followed. Did the authors check the nature of the distribution of features? With what method? Some features, such as relative expression, cannot initially be attributed to natural numbers, that is, they cannot be compared by criteria that require a normal distribution.
Reviewer 4 Report
The manuscript entitled "Leptin-Induced HLA-G inhibits myometrial contraction and differentiation" develops a topic of considerable interest.
The title and purpose are consistent with what was done in the study.
The MMs are well explained and the methods used are consistent with what the authors wanted to investigate.
The results are expressed in a clear and understandable way by the reader. The graphs and figures are clear and help to understand the results.
The discussion is well set up and the deductions reported are logical and explain the effects obtained.
The conclusions are in accordance with the title and purpose.
Therefore the work is complete and provides several insights to further expand the knowledge.
Author Response
We thank the Reviewer 4 for his/her positive comments on our manuscript. In particular, we appreciate the Reviewer’s recognition that this work is important in offering new insights on the mechanisms involved in preterm labour.
Round 2
Reviewer 1 Report
Dear Authors,
please find my review in the attached file.
With best regards

Reviewer 3 Report
All comments were answered satisfactorily.
Author Response
We thank the Reviewer for his/her methodological comments on our manuscript. In particular, we appreciate the Reviewer’s recognition of the use of varied and modern methods.